# Use of Drug Claims Data and a Medication Risk Score to Assess the Impact of CYP2D6 Drug Interactions among Opioid Users on Healthcare Costs

**DOI:** 10.3390/jpm11111174

**Published:** 2021-11-10

**Authors:** Veronique Michaud, Ravil Bikmetov, Matt K. Smith, Pamela Dow, Lucy I. Darakjian, Malavika Deodhar, Brian Cicali, Kevin T. Bain, Jacques Turgeon

**Affiliations:** 1Tabula Rasa HealthCare (TRHC), Precision Pharmacotherapy Research and Development Institute, Orlando, FL 32827, USA; vmichaud@trhc.com (V.M.); rbikmetov@trhc.com (R.B.); mksmith@trhc.com (M.K.S.); pdow@trhc.com (P.D.); ldarakjian@trhc.com (L.I.D.); mdeodhar@trhc.com (M.D.); 2Faculty de Pharmacie, Université de Montréal, Montréal, QC H3T 1J4, Canada; 3Center for Pharmacometrics and Systems Pharmacology, Department of Pharmaceutics, College of Pharmacy, University of Florida, Orlando, FL 32611, USA; bcicali@ufl.edu; 4Biophilia Partners, Philadelphia, PA 19103, USA; kbain0225@gmail.com

**Keywords:** CYP2D6, drug-drug interactions, opioids, medical expenditure, pharmacoeconomics, medication risk score

## Abstract

Cytochrome P450 2D6 (CYP2D6) activity is highly variable due to several factors, including genetic polymorphisms and drug-drug-gene interactions. Hydrocodone, oxycodone, codeine, and tramadol the most commonly prescribed CYP2D6-activated opioids for pain. However, the co-administration of CYP2D6 interacting drugs can modulate CYP2D6-medicated activation of these opioids, affecting drug analgesia, effectiveness, and safety, and can impact healthcare costs. A retrospective, observational cohort analysis was performed in a large (*n* = 50,843) adult population. This study used drug claims data to derive medication risk scores and matching propensity scores to estimate the effects of opioid use and drug-drug interactions (DDIs) on medical expenditures. 4088 individuals were identified as opioid users; 95% of those were prescribed CYP2D6-activated opioids. Among those, 15% were identified as being at risk for DDIs. Opioid users had a significant increase in yearly medical expenditure compared to non-opioid users ($2457 vs. $1210). In matched individuals, average healthcare expenditures were higher for opioid users with DDIs compared to those without DDIs ($7841 vs. $5625). The derived medication risk score was higher in CYP2D6 opioid users with interacting drug(s) compared to no DDI (15 vs. 12). Higher costs associated with CYP2D6 opioid use under DDI conditions suggest inadequate CYP2D6 opioid prescribing practices. Efforts to improve chronic opioid use in adults should reduce interacting drug combinations, especially among patients using CYP2D6 activated opioids.

## 1. Introduction

Chronic pain is prevalent in the U.S. population [1,2]. Studies conducted over the past two decades have estimated the prevalence of chronic pain to range from 10% to 60%, and sometimes as high as 80%, among adults (aged ≥ 18 years) [3,4,5,6,7]. Chronic pain is defined as pain that persists beyond the expected healing time and is ongoing, lasting at least three to six months [8,9]. Chronic pain contributes to rising healthcare costs and is linked to a number of physical and mental conditions that result in a loss of productivity [10,11,12]. Better and more effective pain management strategies for chronic pain are needed. Chronic pain is one of the most common reasons adults seek medical care, accounting for 15 to 20% of physician visits, and can lead to a dependence on opioids, poor health, and a reduced quality of life [1,13,14]. In 2017, more than 191 million opioid prescriptions were filled in the U.S., and prescriptions for opioids to treat chronic pain continue to rise dramatically [15,16,17,18].

Several opioids are metabolized by the cytochrome P450 (CYP) enzymatic system, including codeine, fentanyl, hydrocodone, methadone, oxycodone, and tramadol [19,20,21]. Among these opioids, some behave like prodrugs, requiring CYP-mediated metabolism for activation. Specifically, codeine, hydrocodone, oxycodone, and tramadol are metabolized by the polymorphic CYP2D6 isoenzyme into their respective active metabolites, i.e., morphine, hydromorphone, oxymorphone, and O-desmethyl-tramadol [19,20,22,23]. These metabolites are much more potent antagonists of the µ-opioid receptor than their parent compounds, and are primarily responsible for analgesic response. However, the parent compounds may be responsible for -receptor independent adverse effects [20,22,24]. Concomitant administration of CYP2D6 inhibitors or of substrates with greater affinity for the CYP2D6 isoenzyme can interfere with the bioactivation process of these drugs. Consequently, drug-drug interactions (DDIs) involving CYP2D6 opioids are associated with decreased concentrations of opioid active metabolites and may lead to inadequate analgesia [25].

DDIs involving opioids may occur more commonly in clinical practice than generally recognized. More than 65% of patients taking an opioid for chronic pain take at least one other drug concomitantly [25]. In clinical practice, concurrent use of non-opioid drugs increases the risk for patients taking opioids to experience DDIs; these prescribing practices may contribute to the economic and physical burdens associated with chronic pain [26]. Approximately 9.5 to 11.5 million patients are prescribed opioids for chronic pain, and the degree and extent to which these patients might have conditions that put them at risk for potential opioid-involved DDIs has been insufficiently studied [15].

Drug claims data could represent a reliable source to attribute risk of adverse drug events (ADE) associated with medications in outpatient populations [27]. We recently reported the association of a proprietary medication risk score (MRS) based on drug claims with health outcomes including ADE, medical expenditures, hospitalizations, emergency department visits, hospital length of stay and death [28,29,30].

In this study, we conducted a retrospective, observational cohort analysis of a large adult population using drug claims data. Our objectives were to (1) describe and quantify the use of opioids and concomitant drugs known to interfere with opioid metabolism, (2) estimate healthcare costs associated with DDIs among patients using opioids (compared to non-opioid users or opioid users without CYP2D6 DDIs), and (3) investigate the impact on the MRS. We hypothesized that prescribing CYP2D6 opioids with the potential for DDIs is associated with higher healthcare costs compared to prescribing non-opioids or CYP2D6 opioids without DDIs and, concurrent increased MRS values. The results of our study quantify prescribing practices for CYP2D6 opioids and highlight both their economic impact and public health implications.

## 2. Materials and Methods

This study utilized Class 4 data consisting of de-identified pharmacy prescription drug claims data (1 October 2016 to 31 December 2016) and one year of de-identified medical expenditure data (1 January 2016 to 31 December 2016) obtained from a private healthcare benefits consultant. Subjects were excluded from the analyses if there were no pharmacy claims in the period analyzed. Data elements analyzed were prescribed drugs, doses, age, and gender for all included drug claims. For data protection, date of birth was represented as a year value, with ages over 89 fixed at 89. All individual-level data were anonymized before being made available for analysis in this study. This research protocol was reviewed and approved by Biomedical Research Alliance of New York Institutional Review Board (BRANY IRB), an independent review board, prior to study initiation and a waiver of authorization to use protected health information was granted (protocol #19-12-132-427, sponsor ID BSG-OPIOID-001). Permission for publication was obtained from the healthcare benefits consultant.

### 2.1. Medication Risk Score

A medication risk stratification was performed. Tabula Rasa HealthCare (TRHC, Moorestown, NJ, USA)) has developed a proprietary medication risk score (MRS, the MedWise^®^ Risk Score) using algorithms that consider five medication characteristics to compute risk of ADEs [28,31]. Briefly, it includes (1) computation of a drug regimen relative odds ratio for adverse drug events using the U.S. Food and Drug Administration pharmacovigilance database (FAERS), (2) anticholinergic cognitive burden, (3) sedative burden (4), drug–induced Long QT Syndrome (LQTS) burden, and (5) CYP450 drug interaction burden risk scores. The total MRS was divided into Minimal, Low, Intermediate, High, and Severe risk sub-categories. The methodology has been recently published and is extensively described in Patents #WO2019089725 and #WO2017213825 [31].

### 2.2. Opioid Identification

Classification of the population into different subgroups based on opioid drug claims has been determined as follows: (1) subjects using drugs other than opioids vs. all opioid using subjects, and (2) CYP2D6 opioid users with no DDIs vs. CYP2D6 opioid users with interacting drugs. Opioid users were defined as individuals who filled a prescription for an opioid in the last three months of 2016. FDA-approved opioid medications included buprenorphine, codeine, fentanyl, hydrocodone, hydromorphone, meperidine, methadone, morphine, oxycodone, oxymorphone, pentazocine, propoxyphene, dextropropoxyphene, tramadol, and tapentadol. Drugs administered by the intravenous or epidural routes were excluded. The prescribed opioids identified among the study cohort are listed on the Figure 1. CYP2D6 metabolized opioids included codeine, hydrocodone, oxycodone, and tramadol. CYP2D6 opioid users with interacting drugs were defined as CYP2D6 opioid users exposed to at least one potential pharmacokinetic drug interaction (including CYP2D6 drug inhibitors or CYP2D6 higher affinity substrates). The identification of CYP2D6 inhibitors and CYP2D6 substrates was based on their drug metabolism pharmacokinetic parameters including their affinity for CYP2D6 isoenzyme and the percentage of their elimination pathway via CYP2D6 (e.g., Km, IC50, intrinsic clearance, clinical drug interaction studies, in vitro and in vivo drug metabolism studies). CYP2D6 substrates were classified into 3 categories based on their affinity towards the CYP2D6 isoenzyme, which determined the relative risk of competitive inhibition between substrates of the same isoenzyme. CYP2D6 metabolized opioids exhibit a weak affinity for CYP2D6, so potential drug-drug interactions were considered clinically significant if concomitant CYP2D6 substrates exhibit higher affinity (i.e., strong and intermediate affinity). CYP2D6 substrates with high and intermediate affinities observed in this study are listed in Appendix A. It should be noted that some members of the population were prescribed more than one opioid, which can be observed in discrepancies between the drug count and population data.

### 2.3. Data Processing and Statistical Analyses

Descriptive population characteristics including comorbidities, age, gender, MRS, and individual risk factors were measured, including means, medians, standard deviations, range, confidence intervals, and proportions as appropriate. The average total daily dosages of opioids per patient were calculated (with consideration of reversed claims) using the USA Centers for Medicare and Medicaid Services (CMS) guidelines and limitation of double maximum FDA approved daily dosage [28,32,33,34].

Comorbidities were derived using National Drug Codes (NDC) obtained from drug claims and converted to substance level RxNorm Concept Unique Identifier (RxCUI) and Anatomical Therapeutic Chemical (ATC) codes sequentially. The resultant ATC codes were used as a proxy to generate 27 potential comorbidity categories based on ATC codes as described by Pratt et al. (pain category being excluded) [35]. Inclusive and exclusive combinations of ATC codes were used to derive certain comorbidities (e.g., hypertension, congestive heart failure) [35]. In addition, administration route and dosage of drugs were considered to derive the following comorbidities: antiplatelets, arrythmia, chronic airway disease, epilepsy, glaucoma, malignancies, transplant.

To perform the medication risk stratification, a webservice interface and customized scripts were used. Medication risk scores were generated by processing prescribed drug claims using NDCs as drug identifiers. Medication data were extracted from the claims and cleaned of errors and inconsistencies through quality and integrity analyses. Since NDCs can also denote non-medications (e.g., medical devices), active medication data was further filtered to exclude these NDCs. Active medication data for each subject was filtered based on prescription dates and days of supply, including any possible refills.

Data are reported as mean ± standard deviation (SD) or median and interquartile range (IQR) for continuous variables. Comparisons among groups were performed using the unpaired Student’s *t*-test. A continuous propensity score (PS) analysis was performed to adjust for inter-group clinical differences. The explanatory variables in the logistic regression analysis performed to generate a PS for each patient (representing the likelihood of being in the interest group) included age, gender, and all comorbidities, excluding inflammatory and pain syndromes. The continuous variable age was checked for the assumption of linearity in the logit. Graphical representations suggested a node at age 45 to split the variable into two linear relationships: one equal to age for values up to age of 45 and 0 after and the second equal to age for values above 45 and zero before. The variables were selected only if they maximized the within-sample correct prediction rates. Interactions between variables were allowed only if they were supported clinically and statistically (*p* < 0.20).

The goodness-of-fit of the model was evaluated using the Hosmer–Lemeshow test. Patients in the interest group were matched 1:1 with patients of the other group based on PS using the greedy matching algorithm with replacement. This approach matches patients on decreasing levels of precision on their PS, beginning with a precision at 6 digits and repeating the process until matches were completed on the 1 digit of the propensity score (13). Given disparities in the number of patients in each group, many patients in the interest group could be suitably matched to more than 1 candidate from the other group. Rather than report results from only 1 random sample of matched pairs of subjects, which may introduce bias, a bootstrap method was performed for each comparison in order to reflect the many possible matched pairs. The estimate generated represents a ‘mean’ with its 95% confidence interval (CI) based on percentiles at 2.5% and 97.5% from the bootstrap distribution obtained from a series of 1000 iterations of Monte-Carlo simulations with replacement. For all variables, according to the central limit theorem, the shape of the sampling distribution from the bootstraps was nearly normal. Statistical significance was considered present when the 95% CI around the bootstrap mean difference did not include zero. Standardized mean difference (SMD) was provided to examine the balance of covariate distributions between groups after the propensity score matching. The SMDs below 0.10 were achieved for almost all covariates.

Analyses were performed using the statistical software SAS version 9.4 (SAS Institute Inc., Cary, NC, USA); Python 3.8.5 using the NumPy (v. 1.19.2), pandas (v. 1.2.1), statsmodels (v. 0.12.1), scikit-learn (v. 0.23.2), Matplotlib (v. 3.3.2), and seaborn (v. 0.11.1) packages; and in R, (v. 1.2.5019) with the dplyr, data.table, sqldf, scales, and ggplot2 packages. Microsoft SQL Server (v. 15) was used to manipulate and analyze large datasets.

## 3. Results

### 3.1. Overall Opioid Usage

In our study, a total of 307,266 drug claims from 50,843 patients were available for the period of 1 October 2016 to 31 December 2016. Characteristics for non-opioid users were compared with characteristics for non-opioid users in Table 1. According to pharmacy claims data, 4088 individuals (8.0%) were opioid users, including 355 subjects (8.7%) who were prescribed more than one opioid concurrently. Opioid users were older and received a higher number of prescribed medications compared to non-opioid users. The 25 most commonly prescribed medications are provided in Appendix A. Drug claims were used to derive drug classes and comorbidities as previously described [35]. Using drugs as a proxy, a substantial difference was observed among opioid vs. non-opioid users in the prevalence of individuals having anxiety (8.00 vs. 3.53%), having depressive disorders (18.03 vs. 16.80%), having gastroesophageal reflux disease (GERD; 12.4 vs. 9.09%), in need of co-prescribed anti-epileptic drugs (12.62 vs. 4.86%), and co-prescribed non-steroidal anti-inflammatory drugs (NSAIDs; 17.44 vs. 4.91%), respectively (*p* = 0.04–0.001). Several of these medications could be considered as part of the pain management strategy or could be related to poor pain management. As listed in Table 2, the most prevalent opioid medications prescribed in our study population were hydrocodone, oxycodone, tramadol, codeine, morphine, and buprenorphine (from 43.5% to 3.2%, respectively).

The total MRS was significantly higher in subjects with a prescribed opioid compared to the non-opioid medication users, with a difference of 4.5 MRS units (95% CI 4.4–4.6) (Table 1). Figure 2a illustrates the adjusted MRS distribution observed in subjects without an opioid medication and in subjects treated with an opioid drug (median MRS of 2 and 7 respectively). The opioid user group was also associated with an increase in the CYP450 drug interaction burden (Table 1). Among the opioid user group, fewer individuals have a Minimal MRS category level compared to non-opioid users (*p* < 0.05). In contrast, higher frequencies of individuals having an MRS categorized as Low, Intermediate, High, and Severe were observed in the opioid user group (Figure 3a,b; all *p* < 0.05).

The impact of opioid medications on yearly medical expenditures is presented in Table 3. Medical expenditures were 2.19-fold higher among individuals receiving opioid medications than non-opioid users: the adjusted mean of total medical expenditures was $2457 vs. $1120 for the opioid vs. non-opioid user groups, respectively. Consistent estimation of the zero-inflated count model indicated that annual medical expenditures for individuals receiving opioid medications were 2.39-fold higher than those who did not take opioid medication ($3912 vs. $1635, respectively).

### 3.2. CYP2D6 Opioids with and without Interacting Drugs

Among all opioid users, 3876 (94.8%) were identified as CYP2D6 opioid users. Characteristics of individuals receiving CYP2D6 activated opioid without and with interacting CYP2D6 medications are reported in Table 4. Of the CYP2D6 opioid users, 577 (15%) were identified as CYP2D6 opioid users with potential DDIs; of these, 100 (17%) had more than one interacting drug. Overall, subjects identified as CYP2D6 opioid users with interacting drugs were approximately 10 years older and were more likely to be female than subjects in the other study group. The average number of prescribed drugs per subject was higher in the group with CYP2D6 interacting drugs (8.0 vs. 4.3 drugs/patient/day). The 25 most prescribed medications in patients receiving CYP2D6 opioids are listed in the Appendix A. Among CYP2D6 opioid users, the presence of CYP2D6 interacting drugs was associated with substantial differences in the prevalence of individuals having anxiety (14.21 vs. 6.49%), depressive disorders (56.85 vs. 10.28%), anti-epileptic drugs (27.38 vs. 9.40%), and GERD (27.90 vs. 9.85%) (*p* < 0.001).

The drug regimen of individuals receiving CYP2D6 opioids with interacting co-medications was associated with a 3.2 unit increase in total MRS when compared to CYP2D6 opioid users without interacting co-medications. (Table 2 and Figure 2b) The medication risk stratification showed that a greater proportion of CYP2D6 opioid users with interacting drugs were ranked in the Low, Intermediate, and High MRS levels compared with those without CYP2D6 DDIs. (Figure 3c,d), *p* < 0.05).

Hydrocodone and oxycodone were the most prescribed CYP2D6 opioids in both groups (with or without DDIs) and accounted for >70% of opioid prescriptions, followed by codeine and tramadol. (Table 2) The five most frequent concomitantly prescribed drugs involved with CYP2D6 DDIs were, in rank order, duloxetine (21.5%, *n* = 124), bupropion (18.7%, *n* = 108), fluoxetine (15.4%, *n* = 89), carvedilol (6.2%, *n* = 36), and paroxetine (6.2%, *n* = 36), as shown in Appendix A. Total percentage exceeds 100% as some individuals were prescribed more than one CYP2D6 interacting drug.

Table 5 reports on the economic analysis performed for annual medical expenditures in subjects receiving CYP2D6 opioids and the impact on these costs if subjects were also receiving a CYP2D6 interacting medication(s). On average, total median expenditures were 2.7- (without DDIs) and 3.1-times (with CYP2D6 DDIs) higher in CYP2D6 opioid users compared to non-opioid users, respectively. Among CYP2D6 opioid users, the presence of interacting co-medications was associated with a 1.4-fold increase (95% CI 1.20–1.62) in medical costs as compared to opioid users without a CYP2D6 interacting co-medication. The analysis using the zero-inflated model confirmed that yearly medical expenditures per patient were significantly higher in the presence of CYP2D6 interacting co-medications among CYP2D6 opioid users ($8030 vs. $6994).

The impact of CYP2D6 DDIs on the CYP2D6 opioid prescribing was assessed using the morphine milligram equivalent (MME) dose (patient-matched analyses). As shown in Table 6, the mean MME daily doses observed for all CYP2D6 opioids were higher among opioid users with CYP2D6 DDIs. On average, the presence of CYP2D6 DDIs was associated with a total MME daily dose of 7.4 ± 48 mg vs. 5.6 ± 32 mg in the group without CYP2D6 DDI.

Our study results estimate the potential for drug-drug interactions and their economic burden to further quantify prescribing practices of opioids in an adult population. In this study, several important findings regarding opioid prescriptions are underscored. First, prescription opioid use is associated with higher MRS, higher healthcare costs, and higher prevalence of comorbidities which could be related to pain syndromes or pain management. Second, the presence of CYP2D6 interacting drugs contributes to additional increases in MRS, higher medical expenditures, and higher total MME daily dose when compared to CYP2D6 opioid users with no DDIs.

The MRS has been previously investigated as a medication risk prediction tool for ADEs and medical outcomes. The MRS was significantly associated with an increase of ADEs, emergency visits and medical expenditures [28]. Recently, a longitudinal study including 427,103 patients showed that the MRS was also independently associated with premature death [29]. Overall, opioid users and CYP2D6 opioid users were associated with higher MRS, indicating that they are at increased risk for ADEs. Based on the MRS, higher healthcare costs observed among opioid users was expected. One of the contributing factors to the MRS is the CYP450 drug interaction burden. Our results demonstrated that opioid users had higher CYP450 drug interaction burden score than the non-opioid group, and the difference was even more pronounced among CYP2D6 opioid users with DDIs vs. those with no DDIs.

Our study results showed that prescription opioid use was prevalent in 8% of this total population. These results show a greater prevalence of opioid use compared to 2011–2012 data from the National Health and Nutrition Examination Survey, where opioid analgesic use was 6.9% in patients aged 20 years and older, an increase from 4.2% in 1999–2002 [16]. In addition, more than 94.8% of patients taking opioids were prescribed opioids metabolized by CYP2D6. Our results are similar to Pergolizzi et al., where the more frequently prescribed opioids were hydrocodone (43%), tramadol, (27%), and oxycodone (15%), as these frequencies compared well with data observed in our study, at 44%, 23%, and 16% respectively [36].

The impact of CYP2D6 activated opioids is important to explore in this population, as CYP2D6 metabolizes these opioids into potent metabolites, which are primarily responsible for their analgesic response [21,37,38]. Among the CYP2D6 opioids users, a substantial proportion (15%) were identified as being exposed to potential DDIs involving their opioids. As stated in the results section, the most concomitantly prescribed drugs that interact with CYP2D6 opioids are duloxetine, bupropion, fluoxetine, carvedilol, and paroxetine. Though some of these drugs (e.g., paroxetine, fluoxetine, bupropion and duloxetine) could be utilized in the treatment of depressive disorders, it is possible that these drugs are being prescribed to aid in pain-related conditions in conjunction with opioid use [39,40]. However, the concomitant use of such drugs exhibiting a stronger affinity for the CYP2D6 enzyme may interfere with the active metabolite formation from the CYP2D6 opioids, which may impede the analgesic efficacy. In fact, multiple studies have shown treatment failure when paroxetine or fluoxetine drugs were used in conjunction [41,42,43,44,45].

In this study, it was observed that 15% of CYP2D6 opioid users were exposed to significant CYP2D6 DDIs. Consistent with this finding, a study reported that the presence of CYP450 drug–drug exposure was common among chronic back pain patients on long term CYP450-metabolized opioids [36]. The overall prevalence of CYP450 drug–drug exposure was 27% and women were more likely to have CYP450 drug–drug exposure compared to males [36]. Similar observations were reported in patients with osteoarthritis taking CYP450-metabolized opioids [46]. In agreement with these findings, we observed a higher percentage of females vs. males taking CYP2D6 opioids and exposed to DDI in our population (58.6 vs. 41.1%, respectively). The studies from Pergolizzi et al. included opioids metabolized by CYP450 (CYP2D6 and CYP3A4), whereas our study focused on opioids activated by CYP2D6. CYP3A4-metabolized drugs were also considered in their analyses; this can explain the higher prevalence of potential DDI observed among long-term opioid users. In our study population, 31% of CYP2D6 opioid users received concomitant CYP3A4 interacting drugs (including CYP3A4 inhibitors, and drugs exhibiting strong or intermediate affinity). We observed that 15% of CYP2D6 opioid users were exposed to significant DDI while specifically looking at CYP2D6. Consistent with this finding, a study reported that the presence of CYP450 drug–drug exposure was common among chronic back pain patients on long term CYP450-metabolized opioids [36]. In that study, the overall prevalence of CYP450 drug–drug exposure was 27% as they included opioids metabolized by CYP450 (CYP2D6 and CYP3A4). Similar observations were reported in patients with osteoarthritis taking CYP450-metabolized opioids [46]. In our study population, 31% of CYP2D6 opioid users received concomitant CYP3A4 interacting drugs (including CYP3A4 inhibitors, and drugs exhibiting strong or intermediate affinity). Pergolizzi et al. also observed that woman were more likely exposed to CYP450 drug-drug interactions compared to male [36]. In agreement with these findings, we observed a higher percentage of females taking CYP2D6 opioids and exposed to DDI in our population (58.6 vs. 41.1%, respectively).

The relationship between patient outcomes and drug-drug interactions in patients taking CYP2D6-metabolized opioids needs further investigation, as it suggests that patients could be experiencing poor pain control or insufficient management of pain related comorbidities. Our study showed that patients with DDIs received higher MME daily doses than CYP2D6 opioid users without DDIs. This finding suggests that the presence of DDI may increase risk of treatment failure by decreasing conversion of CYP2D6-activated opioids to active metabolites leading to increasing the dose of opioids unnecessarily. This could then foster the opioid cascade and co-analgesic requirements.

It was estimated that the economic cost of pain in the U.S. ranges from $560 to $635 billion annually in medical cost and productivity [47]. Our study demonstrated that the use of prescribed opioids is associated with significant increases in medical expenditure, estimated to be a 2.2-fold extra cost. In this population, the yearly average total medical expenditure in individuals receiving CYP2D6 opioid with and without CYP2D6 interacting co-prescribed drugs was estimated at $7841 and $5625 compared to $2368 for matched-non-opioid users; these cost estimates correspond to an annual extra-cost of $5473 and $3257, respectively. Further, in adults prescribed a CYP2D6-metabolized opioid medication, the presence of CYP2D6 DDIs was associated with an increased cost of 1.4-fold annually, equivalent to $2216. Pergolozzi et al. assessed the economic impact of CYP450 DDIs in patients with low chronic back pain taking opioids [25]. They reported similar trends in both younger and older patients with DDIs with significant higher total payments (including medical and prescription) than no-DDI patients [25]. The presence of DDIs resulted in a total payment differential of $733 for younger and $763 for older patients for a six month period [25]. Similar to our study, their results correspond to a 1.11-fold difference in total cost due to DDIs in chronic low back pain patients taking CYP450-metabolized opioids.

This study has potential limitations. First, the results of this study do not necessarily represent a causal relationship, and therefore are considered a statistical association only. Adherence, prescription cost, and disease codes were not available, and thus were not accounted for in this study. Data were obtained from an employee benefit health plan providing services in several states nationwide in the US. Coverage can differ from state to state, however members included in this study were from various states and covered by the same health insurer; this condition should minimize bias. Although some patients can be also enrolled in Medicare and/or Medicaid programs, our findings are mainly applicable to commercially insured patients. Further investigations are required to extrapolate our results to Medicare/Medicaid insured patients. Many previous drug interaction studies on opioids involved small, specific populations taking certain drugs with opioids. This study demonstrated, through observation, that even in a relatively large patient population using a wide variety of drugs, DDIs result in greater associated healthcare costs. Healthcare utilization costs in response to opioid prescription and use is expensive for patients, providers, and insurance companies. Therefore, any reasonable efforts to reduce costs without compromising patient care should be considered, including the use of clinical decision support systems. Though this study is limited in understanding the costs associated with healthcare utilization and CYP2D6 opioids, the quantification of prescribed opioids and of prescribed of opioids with a concomitant interacting drug is important to consider. By using the MRS to inform medication optimization, costs can be reduced. In many cases, we foresee that changing the time of administration of competing drugs or prescribing alternate drugs, such as non-CYP2D6 opioid or non-opioid analgesics (that do not interact at the CYP2D6 enzyme), could reduce drug interaction burden, improve health outcomes, and reduce costs.

## 4. Conclusions

In conclusion, this study reveals the prevalence of DDIs in a large population of patients treated with CYP2D6-activated opioids. Our results demonstrated that opioid use is associated with economic burden and that higher costs are observed when CYP2D6-activated opioids are used concomitantly with other CYP2D6 interacting drugs. These observations are indicative of inadequate CYP2D6 opioid prescribing practices in a real-world population. These findings also suggest avoiding the use of chronic opioids with interacting drug combinations, especially among patients using CYP2D6 metabolized opioids.

## Figures and Tables

**Figure 1 jpm-11-01174-f001:**
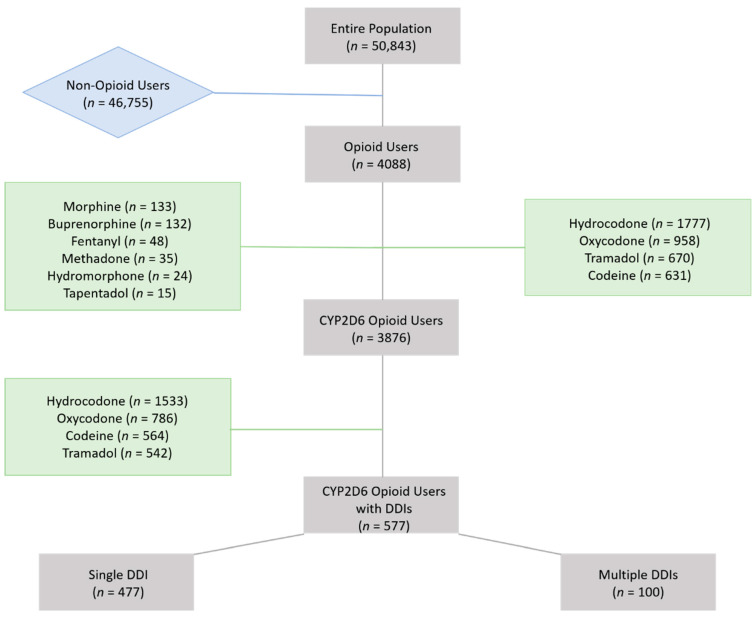
Identification of the study cohort and the subgroups based on drug claims.

**Figure 2 jpm-11-01174-f002:**
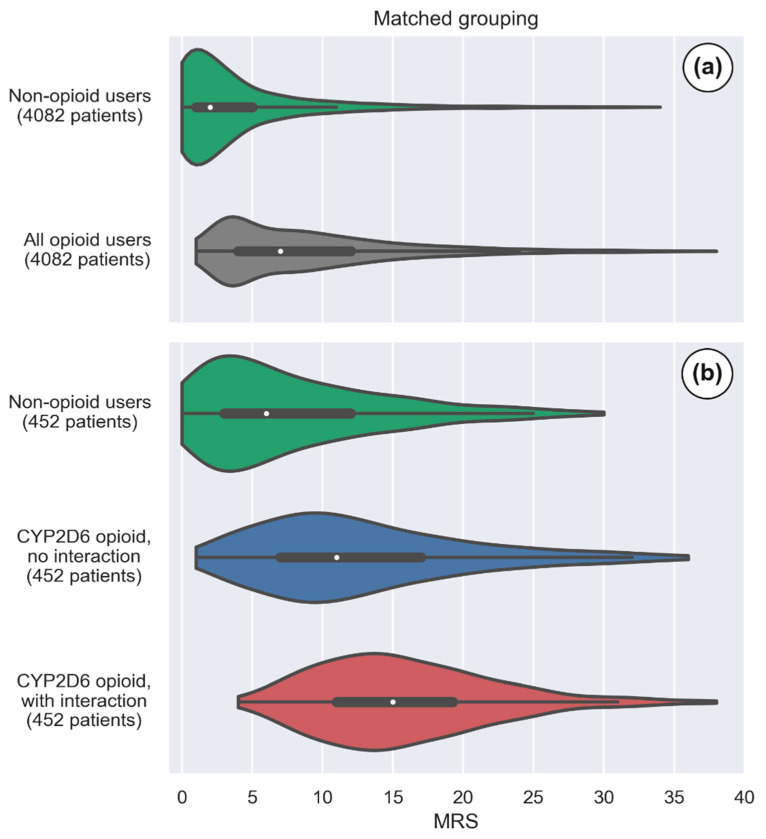
Violin plots of the medication risk score (MRS). Panel (**a**) shows the adjusted MRS distribution in non-opioid and opioid users (with a 1:1 matching as described in the Data Processing and Statistical Analyses section) in the overall population (green and gray, respectively). Panel (**b**) shows the adjusted MRS distribution among CYP2D6 activated opioid users without and with CYP2D6 interacting co-prescribed drugs (blue and red, respectively, and again with a 1:1 matching). A matched non-opioid user group is illustrated in green. The white dots are the medians, and the colored areas are probability density estimates. The number of patients in each violin is the number where matching was possible among the specified groups.

**Figure 3 jpm-11-01174-f003:**
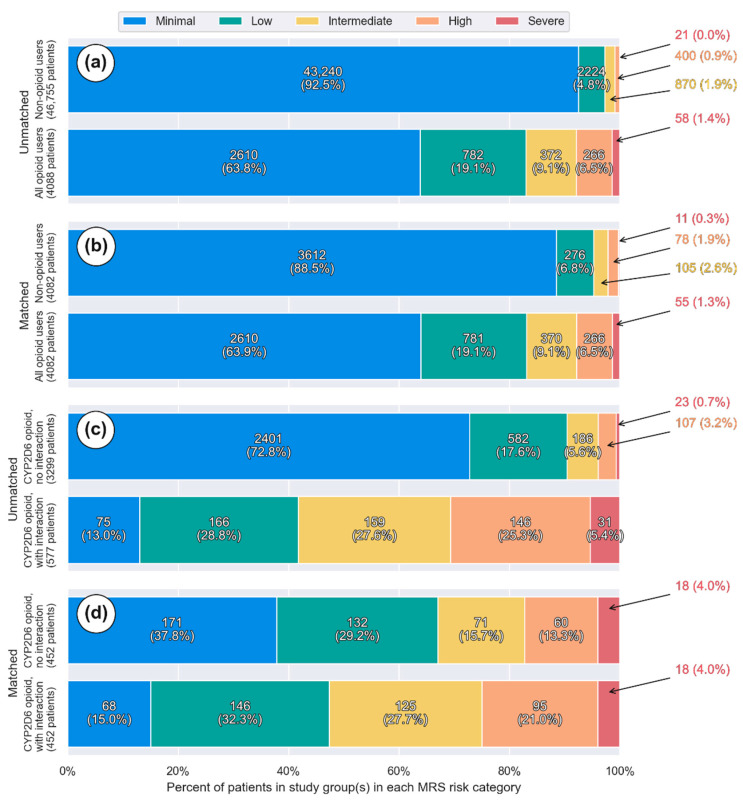
Histogram showing frequency of individuals in Minimal, Low, Intermediate, High, and Severe MRS categories in (**a**) non-opioid user and opioid user groups (unmatched); (**b**) non-opioid user and opioid user groups (matched); (**c**) CYP2D6 activated opioid users without and with CYP2D6 concomitant interacting drugs (unmatched); and (**d**) CYP2D6 activated opioid users without and with CYP2D6 concomitant interacting drugs (matched). The blue, green, yellow, orange, and red represent the Minimal (MRS < 0–9), Low (MRS 10–14), Intermediate (MRS 15–19), High (MRS 20–29) and Severe (MRS ≥ 30) risk categories, respectively.

**Table 1 jpm-11-01174-t001:** Characteristics of the overall population and individuals receiving at least one opioid medication.

*n* Total 50,843	No-Opioid Group	Opioid Group	*p*-Value or Difference
*n* (%)	46,755 (92%)	4088 (8%)	
Age: y ± SD *	40.4 ± 18.5	44.9 ± 14.5	<0.001
Gender:			
Male (%)	19,473 (41.6)	1879 (46.0)	
Female (%)	27,282 (58.4)	2209 (54.0)	<0.001
Number of prescribed drugs per patient: mean ± SD	2.6 ± 2.0	4.8 ±3.0	<0.001
Drug class/co-morbidity (using drug as a proxy): *n* (%)			
Anticoagulants	615 (1.32)	115 (2.81)	<0.001
Antiplatelet drugs	601 (1.29)	76 (1.86)	0.003
Anxiety	1652 (3.53)	327 (8.00)	<0.001
Arrythmia	357 (0.76)	63 (1.54)	<0.001
BPH	628 (1.34)	101 (2.47)	<0.001
Chronic airway disease	5735 (12.27)	421 (10.30)	<0.001
Cardiac heart failure	1836 (3.93)	152 (3.72)	0.53
Dementia	21 (0.04)	1 (0.02)	1.0
Depression	7853 (16.80)	737 (18.03)	0.04
Diabetes	3477 (7.44)	281 (6.87)	0.19
Epilepsy	2271 (4.86)	516 (12.62)	<0.001
GERD	4251 (9.09)	507 (12.4)	<0.001
Glaucoma	666 (1.42)	47 (1.15)	0.16
Gout	587 (1.26)	50 (1.22)	0.94
HIV	103 (0.22)	7 (0.17)	0.72
Hyperlipidemia	7965 (17.04)	618 (15.12)	0.002
Hypertension	7500 (16.04)	685 (16.76)	0.23
Hyperthyroidism	59 (0.13)	2 (0.05)	0.24
Incontinence	304 (0.65)	56 (1.37)	<0.001
NSAIDs	2296 (4.91)	713 (17.44)	<0.001
Malignancies	111 (0.24)	8 (0.20)	0.74
Migraine	626 (1.34)	68 (1.66)	0.09
Parkinson	0	0	
Psoriasis	119 (0.25)	9 (0.22)	0.87
Psychotic illness	544 (1.15)	77 (1.88)	<0.001
Transplant	191 (0.41)	11 (0.27)	0.19
Tuberculosis	3 (0.01)	0	1.0
Total MRS: mean (95% CI) ^ŧ^	3.5 (3.4–3.6)	8.0 (7.9–8.1)	4.5 (4.4–4.6)
CYP450 drug interaction burden score: mean (95% CI) ^ŧ,^**	3.4 (3.3–3.5)	4.5 (4.4–4.5)	1.1 (0.9–1.2)

* missing data for 340 and 4 subjects in the non-opioid and opioid groups, respectively. ^ŧ^ patient-matched analyses. ** zero-inflated model was used. Abbreviation: BPH, benign prostate hyperplasia; GERD, gastroesophageal reflux disease; NSAIDs, nonsteroidal anti-inflammatory drug.

**Table 2 jpm-11-01174-t002:** Most prevalent prescribed opioid medications, and CYP2D6 activated opioids in individuals with and without CYP2D6 interacting medications.

Group	Opioids	*n* (%) *
Overall opioid users (*n* = 4088)	Hydrocodone Oxycodone Tramadol Codeine Morphine Buprenorphine Fentanyl Methadone Hydromorphone Tapentadol	1777 (43.5) 958 (23.4) 670 (16.4) 631 (15.4) 133 (3.3) 132 (3.2) 48 (1.2) 35 (0.9) 24 (0.6) 15 (0.4)
CYP2D6 activated opioid_No interaction (*n* = 3299)	Hydrocodone Oxycodone Codeine Tramadol	1533 (46.5) 786 (23.8) 564 (17.1) 542 (16.4)
CYP2D6 activated opioid_With interacting drug(s) (*n* = 577)	Hydrocodone Oxycodone Tramadol Codeine	244 (42.3) 172 (29.8) 128 (22.2) 67 (11.6)

* Individuals can receive more than one opioid medication.

**Table 3 jpm-11-01174-t003:** Yearly total medical expenditure in the overall insured population without prescribed opioid compared to individuals receiving at least one opioid medication.

*n* = 4082	No-Opioid	Opioid	Fold-Difference
Total medical expenditure: median (95%CI)	$1370 (1293–1447)	$4043 (3907–4178)	
Total medical expenditure: mean (P2.5th- P97.5th) *	$1120 (1061–1184)	$2457 (2369–2548)	2.19 (2.05–2.34)
Zero-inflated model			
Total medical expenditure: mean (P2.5th-P97.5th) *	$1635 (1562–1711)	$3912 (3805–4023)	2.39 (2.26–2.52)

Patient-matched analysis. * Log2 transformed data.

**Table 4 jpm-11-01174-t004:** Characteristics of individuals receiving CYP2D6 activated opioid medication without and with CYP2D6 interacting co-prescribed drugs.

*n* Total 3876	CYP2D6 Activated Opioid_No Interaction	CYP2D6 Activated Opioid_with Interacting Drugs	*p*-Value or Difference
*n* (%)	3299 (85%)	577 (15%)	
Age: y ± SD	43.7 ± 14.8	52.8 ± 10.4	<0.001
Gender			
Male: *n* (%)	1517 (46.0)	239 (41.4)	
Female: *n* (%)	1782 (54.0)	338 (58.6)	0.046
Number of prescribed drugs per patient: mean ± SD	4.3 ± 2.5	8.0 ± 3.4	<0.001
Drug class/co-morbidity (using drug as a proxy): *n* (%)			
Anticoagulants	68 (2.06)	39 (6.76)	<0.001
Antiplatelet drugs	39 (1.18)	32 (5.55)	<0.001
Anxiety	214 (6.49)	82 (14.21)	<0.001
Arrythmia	35 (1.06)	23 (3.99)	<0.001
BPH	71 (2.15)	26 (4.51)	0.002
Chronic airway disease	314 (9.52)	94 (16.29)	<0.001
Cardiac heart failure	16 (0.48)	126 (21.84)	<0.001
Dementia	0 (0)	1 (0.17)	0.15
Depression	339 (10.28)	328 (56.85)	<0.001
Diabetes	171 (5.18)	92 (15.94)	<0.001
Epilepsy	310 (9.40)	158 (27.38)	<0.001
GERD	325 (9.85)	161 (27.90)	<0.001
Glaucoma	26 (0.79)	13 (2.25)	0.005
Gout	35 (1.06)	13 (2.25)	0.024
HIV	6 (0.18)	1 (0.17)	1.0
Hyperlipidemia	379 (11.49)	212 (36.74)	<0.001
Hypertension	432 (13.09)	213 (36.92)	<0.001
Hyperthyroidism	1 (0.03)	0 (0.0)	1.0
Incontinence	35 (1.06)	18 (3.12)	0.001
NSAIDs	583 (17.67)	111 (19.24)	0.38
Malignancies	5 (0.15)	2 (0.35)	0.28
Migraine	43 (1.30)	20 (3.47)	0.0005
Parkinson	0	0	
Psoriasis	4 (0.12)	5 (0.87)	0.005
Psychotic illness	27 (0.82)	35 (6.07)	<0.001
Transplant	5 (0.15)	4 (0.69)	0.033
Tuberculosis	0	0	
Total MRS: mean (95% CI) *	12.4 (12.1–12.8)	15.7 (15.4–15.9)	3.2 (6.9–12.3)
CYP450 drug interaction burden score: mean (95% CI)*	4.5 (4.3–4.6)	6.6 (6.4–6.7)	2.1 (1.9–2.3)

* patient-matched analyses. Abbreviation: BPH, benign prostate hyperplasia; GERD, gastroesophageal reflux disease; NSAIDs, nonsteroidal anti-inflammatory drug.

**Table 5 jpm-11-01174-t005:** Yearly total medical expenditure in individuals receiving CYP2D6 activated opioid medication without and with CYP2D6 interacting co-prescribed drug(s).

*n* = 452	No-Opioid	CYP2D6 Activated Opioid_No Interaction	CYP2D6 Activated Opioid_with Interacting Drugs	Fold-Difference (CYP2D6 Opioid Users No vs. with Interactions)
Total medical expenditure: median (95%CI)	$2938	$7832 (6972–8684)	$9158 (8394–10,011)	
Total medical expenditure: mean (P2.5th- P97.5th) *	$2368 (1977–2833)	$5625 (4961–6421)	$7841 (7247–8459)	1.40 (1.20–1.62)
Zero-inflated model				
Total medical expenditure: mean (P2.5th-P97.5th) *	$3060 (2643–3539)	$6994 (6270–7742)	$8030 (7462–8615)	1.15 (1.01–1.32)

Patient-matched analysis. * Log2 transformed data.

**Table 6 jpm-11-01174-t006:** The mean total daily dosage of CYP2D6 opioids per patient and the corresponding morphine milligram equivalent dose.

CYP2D6 Opioid	CYP2D6 Activated Opioid_No Interaction (*n* = 452)		CYP2D6 Activated Opioid_with CYP2D6 Interacting Drugs (*n* = 452)	
	Total daily dose (mg)	Total daily MME	Total daily dose (mg)	Total daily MME
Codeine	14 ± 191 (6 to 720)	2.1 ± 28.7 (0.9–108)	31 ± 343 (6 to 720)	4.7 ± 51.5 (0.9–108)
Hydrocodone	4 ± 15 (0.5 to 80)	4 ± 15 (0.5–80)	5 ± 34 (0.5 to 80)	5 ± 34 (0.5–80)
Oxycodone	7 ± 28 (0.5 to 180)	10.5 ± 42.0 (0.8–270)	9 ± 48 (2 to 180)	13.5 ± 72.0 (3–270)
Tramadol	21 ± 99 (2 to 409)	2.1 ± 9.9 (0.2–41)	32 ± 96 (5 to 600)	3.2 ± 9.6 (0.5–60)
Total MME *	**5.6 ± 32 (0.2–270)**		**7.4 ± 48 (0.5–270) ****	

Patient-matched analysis. * MME morphine milligram equivalent. ** *p*-value < 0.001.

## Data Availability

Data obtained are kept in a HIPPA, HiTRUST accredited platforms and individual patient’s data obtained by the third party cannot be made publicly available.

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
