# Peer review of "Use of Drug Claims Data and a Medication Risk Score to Assess the Impact of CYP2D6 Drug Interactions among Opioid Users on Healthcare Costs"

_jpm, 2021, doi:10.3390/jpm11111174_

Round 1

Reviewer 1 Report

Authors have conducted a retrospective, observational cohort analysis to estimate the effects of opioid use and DDIs on medical expenditure. However, there are number of concerns (mentioned below) which should be resolved:

  1. Abstract should be clear.
  2. The duration of study is so small.
  3. Inclusion and Exclusion criteria not clear.
  4. FAERS contains duplicates. How author deal with it?
  5. p<0.20?? Give suitable reference for this value.
  6. How stastical tests were applied as number of patients in non-opioid and opioid group is different? Thus, due to differences in numbers, its obvious to get difference between two groups? Please explain.
  7. Which pharmacoeconomic study was done?
  8. Is WTP calculated?
  9. Its not clear after calculating the cost what analysis was further done to make a valid conclusion.
  10. Conclusion need revision.

Author Response

Authors have conducted a retrospective, observational cohort analysis to estimate the effects of opioid use and DDIs on medical expenditure. However, there are number of concerns (mentioned below) which should be resolved:

1. Abstract should be clear.

The abstract was revised as suggested by the reviewer.

2. The duration of study is so small.

Pharmacoeconomic analyses performed using claim data from health care plans are somewhat difficult to carry from one year to the other as members often decide to switch from one plan to the other. Consequently, a lot of data are lost. Second, analyses performed over a one-year period do not require significant adjustments in regards to fluctuations in cost of medical interventions and services provided from one year to the other. It also limits the variability observed in total medical expenditures for each specific member as their total costs tend to increase year-over-year due to the disease-related normal trajectory. We agree with the reviewer that the one-year observation is a snap-shop of the global situation but consider that such information remains of great value.

3. Inclusion and Exclusion criteria not clear.

As described on Figure 1 and section 2.2, patients who received a prescribed opioid were included to compare opioid user vs non-opioid users. Second, patients who received an opioid metabolized by the CYP2D6 isoenzyme, namely, codeine, tramadol, hydrocodone, and oxycodone were identified. Among CYP2D6 opioid users, patients taking CYP2D6 concomitant drugs that may interact with their CYP2D6-metabolized were compared.  For clarity, patient groups are identified on Figure 1.

4. FAERS contains duplicates. How author deal with it?

Before using the extracted raw data from FAERS databases for the corresponding risk score calculation, we preprocess these data using our customized scripts and procedures. By doing so, we account for possible duplicates and inconsistencies in drug names and reported adverse events. Adverse events’ odds ratios are calculated using a 90% confidence interval for statistical significance. The FAERS risk score is computed using a drug regimen relative odds ratio for adverse drug events on preprocessed data.

5. p<0.20?? Give suitable reference for this value.

Mickey RM, Greenland S. The impact of confounder selection criteria on effect estimation. Am J Epidemiol. 1989;129(1):125–37.

Hosmer DW, Lemeshow S. Applied Logistic Regression. New York: Wiley; 2000

Hosmer DW, Lemeshow S. Applied Survival Analysis: Regression Modeling of Time to Event Data. New York: Wiley; 1999.

6. How statistical tests were applied as number of patients in non-opioid and opioid group is different? Thus, due to differences in numbers, its obvious to get difference between two groups? Please explain.

As described in the method section, a continuous propensity score analysis was performed to adjust for inter-group clinical differences. Given disparities in the number of patients in each group, many patients in the interest group could be suitably matched to more than 1 candidate from the other group. Rather than reporting results from only 1 random sample of matched pairs of subjects, which may introduce bias, a bootstrap method was performed for each comparison in order to reflect the many possible matched pairs. Consequently, for all analyses performed, patients were matched. For example, the comparison of the MRS and medical expenditure among opioid vs non-opioid users included the same number of patients in each group (n=4,082 patients in each group). The comparison was not conducted on the entire population, but patients were matched using a continuous propensity score (Figure 1a and Table 3). The same analytical approach was used to investigate the effect of CYP2D6 drug-drug interactions in CYP2D6 opioid users (Figure 1b and Table 5).  In addition, Monte Carlo simulations were conducted to derived 1,000 iterations (test different matches). The statistical analysis performed in this study shall be considered as robust.

Furthermore, the comparison between groups with different sample size does not mean that statistical results will product significant differences. Groups with unequal sample size affect the robustness of the homogeneity of variance assumption. Many statistical articles showed that t-test and ANOVA were robust to moderate departure from this assumption. For instance, Keppel et al. discussed how large the difference between group sample sizes should be for heterogeneity of variance to be a problem; they were unable to define a rule of thumb. Therefore, if equal variances in groups with unequal sample sizes are observed, there is no problem. If unequal variances between groups are observed but sample sizes are nearly equal, there is no problem either. The problem is with unequal variances and unequal sample sizes, statistics that were not observed in our data.

Keppel, G. and Wickens, T. D. (2004). Design and Analysis: A Researcher's Handbook, 4th Edition. Pearson

7. Which pharmacoeconomic study was done?

See Table 3 and Table 5. Yearly total medical expenditure was compared between 1) the overall insured population without prescribed opioid compared to individuals receiving at least one opioid medication and 2) individuals receiving CYP2D6 activated opioid medication without and with CYP2D6 interacting co-prescribed drug(s). Comparisons among groups were performed using the unpaired Student's t-test. A continuous propensity score (PS) analysis was performed to adjust for inter-group clinical differences. And a test of inflated zeros for Poisson regression models was used to model count data that had an excess of zero counts.

8. Is WTP calculated?

The “willingness to pay” is not applicable in this study. Medical expenditures from health insurance plans were used (Part A and B), but the prescription expenditure (Part D) were not available.

9. It’s not clear after calculating the cost what analysis was further done to make a valid conclusion.

Unfortunately, we disagree with the reviewer. We have developed a study design and performed appropriate analyses that support our conclusions. We are presenting a significant amount of results in support of our conclusions. Furthermore, we have taken good care to discuss the observed results in comparison to other published studies. The novel aspect of our analyses is also described and discussed. Responses to comments #5, 6 and 7 provide additional information.

10. Conclusion need revision.

The conclusion was revised as suggested by the reviewer.

Reviewer 2 Report

This is an interesting paper. The aim of the present study is to estimate healthcare costs associated with DDIs among patients using opioids and to investigate the impact on the MRS. Changing the time of administration of competing drugs or prescribing alternate drugs could reduce drug interaction burden, improve health outcomes, and reduce costs.

Comments:

  1. The title should be concise
  2. Line 22: You should present the meaning of DDIs
  3. Line 85: You should detail what drugs Class 4 means.
  4. Line 128: You should present the meaning of CMS
  5. Figure 2: You should describe how the number of patients was calculated
  6. Line 276: You should remove the first parenthesis from the Figure 3d
  7. You should add description for Figure 3c and 3d in the caption of figure 3
  8. The authors should add the list of abbreviations
  9. Lines 536-539: The link for this reference is inactive.

Author Response

This is an interesting paper. The aim of the present study is to estimate healthcare costs associated with DDIs among patients using opioids and to investigate the impact on the MRS. Changing the time of administration of competing drugs or prescribing alternate drugs could reduce drug interaction burden, improve health outcomes, and reduce costs.

Comments:

1. The title should be concise

The title has been revised: Use of drug claims data and a medication risk score to assess the impact of CYP2D6 drug interactions among opioid users on healthcare costs

2. Line 22: You should present the meaning of DDIs

The meaning of DDI has been presented.

3. Line 85: You should detail what drugs Class 4 means.

Class 4 data consist of pharmacy and medical claims data. The sentence was modified for clarity.

4. Line 128: You should present the meaning of CMS

The meaning of CMS has been presented. It corresponds to the USA Center for Medicare and Medicaid Services

5. Figure 2: You should describe how the number of patients was calculated

Altered caption for Figure 2: Violin plots of the medication risk score (MRS). Panel (a) shows the adjusted MRS distribution in non-opioid and opioid users (with a 1:1 matching as described in the Data Processing and Statistical Analyses section) in the overall population (green and gray, respectively). Panel (b) shows the adjusted MRS distribution among CYP2D6 activated opioid users without and with CYP2D6 interacting co-prescribed drugs (blue and red, respectively, and again with a 1:1 matching). A matched non-opioid user group is illustrated in green. The white dots are the medians, and the colored areas are probability density estimates. The number of patients in each violin is the number where matching was possible among the specified groups.

6. Line 276: You should remove the first parenthesis from the Figure 3d

The first parenthesis has been removed.

7. You should add description for Figure 3c and 3d in the caption of figure 3

Figure 3 new caption: Histogram showing frequency of individuals in Minimal, Low, Intermediate, High, and Severe MRS categories a) in non-opioid user and opioid user groups (unmatched); b) in non-opioid user and opioid user groups (matched); c) in CYP2D6 activated opioid users without and with CYP2D6 concomitant interacting drugs (unmatched); and d) in CYP2D6 activated opioid users without and with CYP2D6 concomitant interacting drugs (matched). The blue, green, yellow, orange, and red represent the Minimal (MRS<0-9), Low (MRS 10-14), Intermediate (MRS 15-19), High (MRS 20-29) and Severe (MRS≥30) risk categories, respectively.

8. The authors should add the list of abbreviations

The following list of abbreviations has been added at the end of the manuscript.

DDI        Drug- Drug interaction

CYP        Cytochrome P450

ADE       Adverse drug events

MRS      Medication risk score

TRHC     Tabula Rasa HealthCare

FAERS    USA FDA Adverse Event Reporting System

LQTS      Long QT syndrome

CMS      Centers for Medicare and Medicaid services

NDC       National Drug Codes

ATC       Anatomical Therapeutic Chemical codes

RxCUI    RXNorm Concept Unique Identifier

SD          Standard deviation

PS           Propensity score

IQR         Interquartile range

CI            Confidence interval

SMD       Standardized mean difference

GERD      Gastroesophageal reflux disease

NSAIDs   Non-steroidal anti-inflammatory drugs

BPH         Benign prostate hyperplasia

MME       Morphine milligram equivalent

9. Lines 536-539: The link for this reference is inactive.

An updated citation has been provided.

U.S. Department of Health and Human Services. Opioid Oral Morphine Milligram Equivalent (MME) Conversion Factors. Available online: https://www.hhs.gov/guidance/sites/default/files/hhs-guidance-documents/Opioid%20Morphine%20EQ%20Conversion%20Factors%20%28vFeb%202018%29.pdf (accessed on 25 October 2021).

Round 2

Reviewer 1 Report

Author has addressed all my comments in the revised manuscript

Author Response

Thank you to the reviewer for their comments pertaining to the content of this manuscript.